# The Role of the Brain-Derived Neurotrophic Factor in Chronic Pain: Links to Central Sensitization and Neuroinflammation

**DOI:** 10.3390/biom14010071

**Published:** 2024-01-05

**Authors:** Huan-Yu Xiong, Jolien Hendrix, Siobhan Schabrun, Arne Wyns, Jente Van Campenhout, Jo Nijs, Andrea Polli

**Affiliations:** 1Pain in Motion Research Group (PAIN), Department of Physiotherapy, Human Physiology and Anatomy, Faculty of Physical Education & Physiotherapy, Vrije Universiteit Brussel, 1090 Brussels, Belgium; huanyu.xiong@vub.be (H.-Y.X.); jolien.hendrix@vub.be (J.H.); arne.wyns@vub.be (A.W.); jente.van.campenhout@vub.be (J.V.C.); andrea.polli@vub.be (A.P.); 2Department of Public Health and Primary Care, Centre for Environment & Health, KU Leuven, 3000 Leuven, Belgium; 3Research Foundation—Flanders (FWO), 1000 Brussels, Belgium; 4The School of Physical Therapy, University of Western Ontario, London, ON N6A 3K7, Canada; sschabru@uwo.ca; 5The Gray Centre for Mobility and Activity, Parkwood Institute, London, ON N6A 4V2, Canada; 6Chronic Pain Rehabilitation, Department of Physical Medicine and Physiotherapy, University Hospital Brussels, 1090 Brussels, Belgium; 7Department of Health and Rehabilitation, Unit of Physiotherapy, Institute of Neuroscience and Physiology, Sahlgrenska Academy, University of Gothenburg, 41390 Göterbog, Sweden

**Keywords:** chronic pain, *BDNF*, central sensitization, neuroinflammation, neuroplasticity, epigenetic modifications, *BDNF* polymorphisms

## Abstract

Chronic pain is sustained, in part, through the intricate process of central sensitization (CS), marked by maladaptive neuroplasticity and neuronal hyperexcitability within central pain pathways. Accumulating evidence suggests that CS is also driven by neuroinflammation in the peripheral and central nervous system. In any chronic disease, the search for perpetuating factors is crucial in identifying therapeutic targets and developing primary preventive strategies. The brain-derived neurotrophic factor (*BDNF*) emerges as a critical regulator of synaptic plasticity, serving as both a neurotransmitter and neuromodulator. Mounting evidence supports *BDNF*’s pro-nociceptive role, spanning from its pain-sensitizing capacity across multiple levels of nociceptive pathways to its intricate involvement in CS and neuroinflammation. Moreover, consistently elevated BDNF levels are observed in various chronic pain disorders. To comprehensively understand the profound impact of *BDNF* in chronic pain, we delve into its key characteristics, focusing on its role in underlying molecular mechanisms contributing to chronic pain. Additionally, we also explore the potential utility of *BDNF* as an objective biomarker for chronic pain. This discussion encompasses emerging therapeutic approaches aimed at modulating *BDNF* expression, offering insights into addressing the intricate complexities of chronic pain.

## 1. Introduction

Chronic pain represents a pervasive global health challenge, causing significant disability and socioeconomic burdens [1,2]. With over 30% of the world’s population grappling with chronic pain, it has emerged as a formidable public health concern, posing substantial challenges for both researchers and clinicians [3,4]. Compounding the issue is the inadequacy of existing drug treatments, which often fall short in terms of efficacy and tolerability, leaving more than half of patients with chronic pain without sufficient relief [5,6,7,8]. Groundbreaking research in neuroscience has led the World Health Organization to recognize chronic pain as a disease characterized by intricate functional and structural changes in the brain, neuroinflammation, and increased sensitivity of the central nervous system (CNS) to nociceptive input—central sensitization (CS) [9,10,11,12,13]. In essence, chronic pain not only represents prolonged acute pain but also involves maladaptive neuroplastic changes and sensitization of the nociceptive pathways in the nervous system, extending beyond a simple pain–damage association [14].

In recent years, intensive research has focused on understanding the biochemical and molecular alterations contributing to chronic pain. One promising avenue of exploration centers around the brain-derived neurotrophic factor (*BDNF*) gene, which influences circulating levels of BDNF, and has been implicated in initiating and/or perpetuating neuronal hyperexcitability, maladaptive neuroplasticity, and disinhibition at different levels of nociceptive pathways [15]. *BDNF* assumes a central role in promoting brain homeostasis and neuronal survival, and also serves as a critical regulator of synaptic plasticity [16]. Despite its crucial role in maintaining normal physiological functions, a less positive perspective emerges concerning chronic pain, where mounting evidence suggests a pro-nociceptive role for *BDNF* [17,18]. Consequently, *BDNF* is increasingly acknowledged as a pivotal perpetuating factor in chronic pain.

Here, we offer an up-to-date exploration of the intricate role of *BDNF* in chronic pain, delving into factors potentially contributing to its pathophysiology, such as CS and neuroinflammation. Additionally, this review discusses the potential of *BDNF* as an objective biomarker and an innovative therapeutic target. The scope ranges from specific interventions aimed at *BDNF* expression to indirect approaches targeting its receptors and signaling pathways, providing a comprehensive overview of the current landscape in chronic pain research and potential therapeutic avenues.

## 2. The Physiological Role of *BDNF*

Since its discovery, *BDNF* has garnered extensive attention as one of the most extensively studied neurotrophins, owing to its multipotent impacts on various physiological and pathological functions within the nervous system. Recognized for its robust protective actions promoting brain homeostasis, neuronal survival, synaptogenesis, plasticity, and cognitive function [16,19,20,21], *BDNF* exhibits activity throughout all stages of development and aging [22,23]. *BDNF* plays a crucial role in initiating compensatory processes that facilitate recovery and/or alleviate chronic adverse effects caused by injury or disease in the nervous system [19]. Notably, knockout mice lacking *BDNF* face challenges in reaching adulthood and, when they do, manifest several sensory impairments [24,25].

Like other neurotrophins, BDNF is initially synthesized as a pre-pro-protein. The pre-protein undergoes rapid cleavage to form pro-BDNF, which then assembles in homodimers [26]. The pro-BDNF can be subsequently cleaved by extracellular proteases at synapses and converted to mature BDNF [27]. BDNF exerts its biological functions via two distinct classes of receptors: the high-affinity tropomyosin receptor kinase B (TrkB) and the low-affinity p75 neurotrophin receptor (p75NTR). Pro-BDNF exhibits a preference for binding to p75NTR, while mature BDNF preferentially binds the TrkB receptor. In general, binding to TrkB receptors allows BDNF to modulate and promote neuronal survival, neuroprotection, and long-term potentiation (LTP)—a form of long-term synaptic plasticity in nociceptive pathways [28]. Conversely, binding to p75NTR receptors may regulate neuronal apoptosis, axonal process pruning, and long-term depression (LTD) [29,30]. The contrasting effects of BDNF/TrkB and BDNF/p75NTR signaling form a delicate “yin-yang” system that finely manipulates neuroplasticity and neuronal excitability [31]. Therefore, maintaining a proper balance between these two forms of BDNF is crucial for optimal brain function. Furthermore, dysregulation of *BDNF* functions has been implicated in some diseases such as depression, chronic pain, and neurodegenerative conditions [32].

The functionality of the central nervous system (CNS) is intricately tied to available *BDNF* expression. Predominantly synthesized and expressed in various neuronal cells of the brain, such as sensory neurons and motor neurons [33,34,35], BDNF is also produced, to a lesser extent, in non-neuronal cells such as glial cells and immune cells [36,37]. Available BDNF was found in different regions of the brain, including the neocortex, pyriform cortex, amygdala, hippocampus, claustrum, thalamus, striatum, hypothalamus, and brainstem [33,38]. In addition, circulating BDNF derives from both peripheral and cerebral sources [39,40,41]. Notably, in human studies, serum [42,43,44] and plasma [45] BDNF levels appear to correlate positively with BDNF levels in the brain. Moreover, findings from animal studies suggest a positive correlation between peripheral blood BDNF levels and its concentration in the cerebral cortex, indicating that fluctuations in peripheral blood BDNF levels may reflect changes in the brain [46,47,48,49,50]. However, recent studies comparing cerebrospinal fluid and serum BDNF levels in patients with Alzheimer’s disease have indicated a lack of correlation [51] and that the levels of BDNF in serum or plasma were found to be significantly higher than those in cerebrospinal fluid, possibly due to peripheral synthesis [52].

## 3. The Role of *BDNF* in Patients with Chronic Pain

### 3.1. The Role of BDNF in Central Sensitization

Chronic pain is known to be associated with CS, a process by which the nociceptive signals of neurons at every level of nociceptive pathways are gradually enhanced [53]. CS is responsible for both hyperalgesia and allodynia. At the cellular level, CS occurs in part as a result of enhanced and more efficient synaptic communication between neurons, which primarily involves the reshaping of neuronal circuits, neuronal hyperexcitability, and a reduction in synaptic inhibition [54,55]. Consequently, pain stems from profound changes within the CNS, which not only amplifies responses to nociceptive inputs but also fails to suppress painful signals [56].

Given its essential role throughout the nervous system, *BDNF* has been implicated in the induction and maintenance of the CS. For example, the activation of BDNF/TrkB signaling has been linked to increased pain signaling mechanisms [57]. BDNF, upon release from the dorsal root ganglia, engages with TrkB receptors located on primary afferent nerve endings and post-synaptic tracts in the spinal cord. This interaction serves to amplify and potentiate ascending sensory signals, contributing to the perpetuation of CS. As expected, pain signaling mechanisms can be reversed through intrathecal administration of TrkB inhibitors, which attenuates nociceptive response [58,59]. It is crucial to recognize that CS involves an activity-dependent increase in the excitability of dorsal horn neurons [54], and *BDNF* contributes to this process by promoting a gradual increase in neuronal excitability and synaptic plasticity in the spinal dorsal horn [60,61,62]. Studies have demonstrated that *BDNF* can be synthesized and expressed in the dorsal horn neurons, and the activation of BDNF/TrkB signaling leads to a sustained increase in neuronal excitability, potentially contributing to allodynia, hyperalgesia, and spontaneous pain in neuropathic pain models characterized by CS [63,64].

Persistent CS has been described as a maladaptive neuroplasticity process in chronic pain [65,66]. *BDNF* can regulate synaptic plasticity in an activity-dependent manner, contributing to LTP [67,68]. LTP involves neuronal adaptation at the presynaptic (e.g., increased ability to produce neurotransmitters) and postsynaptic (e.g., increased ability to bind neurotransmitters to receptors) levels, resulting in enhanced synaptic efficiency and, consequently, an increase in the excitability of neuronal pathways [69,70,71]. The synapses are a critical link in inter-neuronal connections, and an increase in their number can facilitate the transmission of nociceptive signals between neurons, potentially contributing to CS [72]. However, *BDNF* knockout specimens exhibited a decrease in preganglionic synaptic innervation density to sympathetic neurons, suggesting that *BDNF* has the ability to increase synaptic density [22]. In addition, with activation of TrkB receptors, there is a downstream activation of various signaling pathway cascades, including the MEK/mitogen-activated protein kinase (MAPK) pathway [73], phosphatidylinositol 3-kinase/protein kinase B (PI3K/PKB) [74], PI3K/Akt/mammalian target of rapamycin (mTOR) pathway [75], and nuclear factor kappaB (NF-κB) signaling pathways [76]. Each of these processes contributes to the induction and maintenance of CS in different parts of nociceptive pathways by facilitating LTP.

Finally, dysfunction in the descending inhibitory nociceptive modulation pathways emerges as a crucial contributor to CS. Recent studies have unveiled reduced intracortical inhibition in different pain populations compared to healthy subjects, with this reduction being associated with more severe pain symptoms [77,78,79,80]. Disinhibition of GABAergic and glycinergic synaptic transmission in nociceptive circuitry is crucial to the generation of chronic pain. Centrally, *BDNF* can weaken GABAergic inhibitory synapses by reducing the expression of potassium-chloride cotransporter 2 (KCC2), thus suppressing the intrinsic inhibitory circuits [61,81,82]. Furthermore, Caumo W. et al. [78] demonstrated an inverse correlation between serum BDNF levels and conditioned pain modulation (CPM) in patients with chronic musculoskeletal pain, highlighting *BDNF*’s involvement in the impairment of the descending inhibitory nociceptive modulation system. Altered CPM is in fact often observed in individuals with persistent pain [83].

### 3.2. Neuroinflammation Drives Chronic Pain via Glial-Derived BDNF and CS

While acute inflammation is responsible for triggering acute pain sensations, neuroinflammation is supposed to play an important role in the chronification and persistence of pain [84,85]. This neuroinflammation is initiated by the activity-dependent release of glial activators, including neurotransmitters, chemokines, and proteases. This release stems from the central terminals of primary afferent neurons or is prompted by the disruption of the blood–brain barrier. Neuroinflammation is characterized by the activation of glial cells such as microglia and astrocytes, the infiltration of immune cells, vasculature changes, and an increased release of inflammatory and glial mediators like cytokines, chemokines, and BDNF [86]. These glial mediators can significantly regulate both excitatory and inhibitory synaptic transmission, thereby contributing to CS and enhanced chronic pain states. Moreover, glial mediators can further act on glial and immune cells to facilitate neuroinflammation through autocrine and paracrine routes [87].

In the spinal cord and brain, glial cells also produce nerve growth factors and neurotrophins, such as BDNF and basic fibroblast growth factor (bFGF), which can affect neuronal function and may contribute to neurotoxicity in several brain pathologies [19]. In fact, the expression of neurotrophins is often upregulated in chronic inflammatory diseases due to their involvement in energy homeostasis [88]. For example, microglial activation following peripheral nerve injury upregulates purinergic receptors, especially P2 × 4R, leading to p38-MAPK phosphorylation and subsequent BDNF release [89]. This microglial-derived BDNF has been implicated in facilitating neuropathic pain and morphine hyperalgesia [90,91]. Additionally, evidence from animal studies also indicated that the synthesis and release of BDNF were significantly increased during inflammatory pain [92,93], cancer pain [94], and neuropathic pain [95]. In the context of neuroinflammation, evidence points to a pivotal shift in astrocytic behavior. Astrocytes, once adept at maintaining homeostatic concentrations of glutamate (Glu) and potassium (K^+^), undergo a transformation wherein they gain the ability to secrete ATP, Glu, and chemokines [96]. This phenomenon contributes to CS and LTP. Glial-derived BDNF may mediate CS by attenuating inhibitory synaptic transmission. As a pain mediator and modulator, *BDNF* can intricately manipulate excitatory glutamatergic and inhibitory GABAergic/glycinergic signals [97,98]. 

A key player in inflammatory activation is the nuclear factor-kappa B (NF-κB), a transcription factor that triggers the expression of pro- and anti-apoptotic genes [99]. Remarkably, the binding of BDNF to the TrkB receptor serves as a trigger for the induction of the NF-κB expression. Furthermore, chronic inflammatory pain has been reported to induce an upregulation of TrkB mRNA and protein expression in the dorsal horn [100]. An additional layer of complexity arises from a p75NTR-mediated effect on NF-κB expression, as evidence suggests that peripheral inflammation induces an upregulation of pro-BDNF and p75NTR in the spinal cord [101,102]. With the activation of p75NTR, pro-BDNF can activate several downstream signaling pathways, including extracellular signal-regulated kinase (ERK)1 and ERK2, NF-κB, and c-Jun N-terminal kinase (JNK) pathways, further promoting the neuroinflammatory state [103,104,105]. These signaling pathways can trigger a series of changes, including neuronal hyperexcitability, LTP, maladaptive neuroplasticity, and an imbalance in excitatory/inhibitory neurotransmission—all of which are intricately involved in the process of CS (Figure 1). This cycle persists as long as the stressor exists, potentially evolving into a serious chronic pain state. Hence, neuroinflammation may drive chronic pain via CS, which can be induced and maintained by cytokines, chemokines, BDNF, and other glia-produced mediators.

Remarkably, the connection between *BDNF* and neuroinflammation remains an underexplored area of research. The association between neuroinflammation and persistent pain underscores the importance of investigating inflammatory factors, particularly *BDNF*, as promising therapeutic targets for the management of chronic pain.

### 3.3. Pro-Nociceptive and Anti-Nociceptive Role of BDNF

Emerging evidence from human studies has revealed higher cerebrospinal fluid [106], plasma [107,108,109], and serum [110,111,112,113] levels of BDNF in patients with chronic pain compared to healthy individuals, which were positively correlated with more severe pain symptoms (Table 1). For instance, higher serum BDNF levels were associated with lower pressure pain thresholds in patients with fibromyalgia [113]. Notably, a one-month treatment with duloxetine (an antidepressant) not only alleviated pain but also led to reduced serum BDNF levels [114], supporting a pro-nociceptive role of *BDNF* in chronic pain. Recent evidence also supports the pro-nociceptive role of *BDNF* in arthritis pain [115], with higher plasma BDNF levels observed in patients with knee osteoarthritis compared to healthy controls, positively correlating with self-reported pain levels [109]. BDNF and TrkB were identified in nerve fascicles within synovial tissue from both patients with osteoarthritis and animal models of inflammatory arthritis [116,117]. As expected, experimental injection of peripheral BDNF increased pain behavior [117]. In addition, inhibition of BDNF/TrkB signaling in animal models of postherpetic neuralgia attenuated mechanical allodynia, reduced inflammation, and reversed neuronal hyperexcitability [18]. These findings strongly advocate that upregulated *BDNF* expression in chronic pain is not merely a byproduct, but a pivotal causal factor.

Despite previous studies indicating a strong involvement of *BDNF* in the nociceptive system, its precise role remains uncertain. This uncertainty is further compounded by conflicting findings, with some research indicating a potential anti-inflammatory effect [118,119,120]. For instance, preliminary evidence from animal research suggests that the release of BDNF can alleviate allodynia and hyperalgesia induced by chronic constriction injury [121]. Additionally, *BDNF* shows anti-inflammatory effects on the animal brain [122], and experimentally induced inflammation, such as the infusion of IL-1 into the hippocampus, and diminishes *BDNF* transcription capacity [123]. Several explanations may account for its potential analgesic effect. Firstly, *BDNF* is involved in the regulation of neural circuits, and alterations in neural circuitry may affect inflammatory responses. Emerging evidence suggests that *BDNF* can inhibit neuroinflammation and regulate cognitive functions [124]. Secondly, the anti-nociceptive effect of *BDNF* may result from central rather than peripheral actions, as elevated levels of peripheral BDNF have been shown to sensitize primary afferent neurons and promote pain hypersensitivity [125]. Moreover, given the fact that inflammation is accompanied by BDNF release [126], upregulated *BDNF* expression in the CNS may have an anti-inflammatory effect following early pain exposure, suggesting a more significant role in acute rather than chronic pain. Finally, *BDNF*’s neuroprotective properties may indirectly contribute to an anti-inflammatory environment. By promoting the survival of neurons and maintaining overall neuronal health, *BDNF* may reduce the release of inflammatory signals associated with cell damage.

**Table 1 biomolecules-14-00071-t001:** Brain-derived neurotrophic factor levels and association with pain in patients with chronic pain.

Reference	Study Population	Source of BDNF Measurement	Mean ± SD BDNF Values	*p*-Values	Correlation with Pain
Polli et al., 2020 [127]	Chronic fatigue syndrome and comorbid fibromyalgia (n = 28) Healthy controls (n = 26)	Serum	Chronic fatigue syndrome and comorbid fibromyalgia: 17.75 ± 4.48, Healthy controls: 14.89 ± 3.55 (values in ng/mL)	Sig	Higher BDNF levels predicted participants’ symptoms and widespread hyperalgesia
Jasim et al., 2020 [128]	Chronic temporomandibular disorder myalgia (n = 39) Healthy controls (n = 39)	Plasma	Chronic temporomandibular disorder myalgia: 263.33 ± 245.13, Healthy controls: 151.81 ± 125.90 (values in pg/mL)	Sig	Not evaluated
Stefani et al., 2019 [111]	Fibromyalgia (n = 117) Osteoarthritis (n = 88) Chronic tensional-type headache (n = 33) Healthy controls (n = 41)	Serum	Osteoarthritis: 24.85, Fibromyalgia: 38.60, Chronic tensional type headache: 37.22, Healthy controls: 22.85 (values in pg/mL)	Sig	Not evaluated
Jablochkova et al., 2019 [107]	Fibromyalgia (n = 75) Healthy controls (n = 25)	Plasma	Fibromyalgia: 1553.30, Healthy controls: 671.6 (values in pg/mL)	Sig	No correlation
Caumo et al., 2016 [78]	Fibromyalgia (n = 19), osteoarthritis (n = 27), myofascial pain syndrome (n = 54), healthy controls (n = 14)	Serum	Fibromyalgia: 50.78 ± 16.06, Osteoarthritis: 17.91 ± 7.27, Myofascial pain syndrome: 29.28 ± 20.01, Healthy controls: 19.00 ± 8.79 (values in ng/mL)	Sig	Higher BDNF levels were significantly correlated with decreased inhibitory system as assessed through conditioned pain modulation
Deitos et al., 2015 [129]	Central sensitivity syndrome absent of structural pathology (n = 81) Central sensitivity syndrome with persistent nociception (n = 59) Healthy controls (n = 37)	Serum	Central sensitivity syndrome absent of structural pathology: 49.87 ± 31.86, Central sensitivity syndrome with persistent nociception: 20.44 ± 8.30, Healthy controls: 14.09 ± 11.80 (values in ng/mL)	Sig	Not evaluated
Bidari et al., 2022 [114]	Fibromyalgia (n = 53) Non-fibromyalgia chronic nociceptive pain (n = 60)	Serum	No differences between the two groups	Not Sig	Decreasing serum BDNF after treatment with duloxetine was associated with the improvement in the disease severity, depression, and pain level
Ranzolin et al., 2016 [130]	Fibromyalgia (n = 69) Healthy controls (n = 61)	Serum	No differences between the two groups	Not Sig	Not evaluated
Iannuccelli et al., 2022 [131]	Fibromyalgia (n = 40) Healthy controls (n = 40)	Serum	Fibromyalgia: 3.38 ± 2.49, Healthy controls: 8.57 ± 3.65 (values in ng/mL)	Sig	No correlation

BDNF, brain-derived neurotrophic factor.

### 3.4. Genetics and BDNF in Chronic Pain

Over the past decade, the identification of altered BDNF levels in individuals with chronic pain has guided many genetic studies, revealing these alterations to be largely genetically determined. Specifically, mutations in the *BDNF* gene have been found to downregulate its secretion and expression, thereby diminishing its impact on the nervous system [132]. The single-nucleotide polymorphism rs6265 in the *BDNF* gene, located in the 5′-prodomain of immature *BDNF* protein and often referred to as Val66Met [132,133], has emerged as a key player in shaping pain perception and pain-related symptoms [134,135], and is associated with vulnerability to different chronic pain disorders [136,137,138] (Table 2). The Val/Val genotype has been linked to a distinct propensity for fibromyalgia symptoms and increased pain catastrophizing [137]. Similarly, research has established a connection between individuals carrying the Val allele and an increased susceptibility to chronic postsurgical pain [138], as well as a correlation with the severity of depression [139]. In addition, a recent clinical study in cancer survivors revealed that those with the Met/Met genotype of *BDNF* rs6265 reported significantly more severe cancer-related neuropathic pain and fatigue than those with other genotypes [140].

*BDNF*, with its influence on crucial neuronal processes, is subject to complex changes in function due to its polymorphism, particularly in modulating neuroplasticity [68]. Evidence suggests that *BDNF* polymorphisms can serve as predictors for responses to experimental pain stimulation and non-invasive brain stimulation techniques, contributing to large interindividual variability in stimulation effects [141,142]. Furthermore, in cultured hippocampal neurons, the *BDNF* protein carrying the Met variant exhibited lower depolarization-induced secretion [134]. Individuals with one or two copies of the *BDNF* Met allele also appear to exhibit decreased brain plasticity, suggesting that the Met allele may influence mechanisms associated with rectifying dysfunctional circuits involved in the imbalance of excitatory and inhibitory systems in the CNS [78,143,144]. In addition, certain *BDNF* polymorphisms have an effect on specific aspects of brain function such as default mode network connectivity, which is currently considered to be central in the pathogenesis of fibromyalgia [145,146]. In conclusion, while chronic pain does not stem from a singular gene, genetic studies focusing on *BDNF* polymorphisms offer valuable insights into potential connections with pain perception and brain plasticity [147,148,149]. These genetic nuances may impact susceptibility to various chronic pain conditions and influence the severity of pain experienced.

**Table 2 biomolecules-14-00071-t002:** *BDNF* Val66Met Polymorphism in patients with chronic pain.

Reference	Study Population	Tissue	Genotyping Method	Genotype Model (BDNF rs6265)	Main Results
Goto et al., 2023 [140]	Female cancer survivors (n = 393)	Buccal swab	Isolated genomic DNA from buccal cells. The analysis of the *BDNF* genotype involved using the PCR SNP Genotyping assays	Val/Val (n = 258) Val/Met (n = 123) Met/Met (n = 12)	Participants with the Met/Met *BDNF* genotype reported significantly worse cancer-related fatigue and neuropathic pain.
Álvaro et al., 2022 [136]	Fibromyalgia (n = 42)	Blood	RT-PCR	Val/Val (n = 30) Val/Met (n = 12)	Val/Met genotypes showed higher efficiency of the descending pain modulatory system and lower disability due to pain. FM patients carrying the Val/Met *BDNF* genotype presented an increased functional connectivity across the motor and prefrontal cortex in response to acute pain associated with differences in acute pain perception and FM.
Yamada et al., 2021 [150]	Chronic low back pain (n = 107)	Blood	RT-PCR	Val/Val (n = 81) Val/Met (n = 26)	No significant associations between the Val66Met genotypes and pain outcomes.
Camila et al., 2020 [137]	Fibromyalgia (n = 108) Healthy controls (n = 108)	Blood	RT-PCR	Val/Val (n = 87) Val/Met (n = 21)	Val allele was significantly more frequent in patients with FM compared to the healthy controls. The *BDNF* Val/Val homozygotes are a potential genetic risk factor associated with higher scores in the Pain Catastrophizing Scale domains: magnification and rumination in patients with FM.
Reddy et al., 2014 [151]	Chronic abdominal pain (n = 18) Healthy controls (n = 31)	Blood	RT-PCR	Val/Val (n = 13) Met allele (n = 5)	No significant associations observed with regard to *BDNF* genotypes with sleep quality or pain grouping.

*BDNF*, brain-derived neurotrophic factor; RT-PCR, real-time polymerase chain reaction; SNP, single nucleotide polymorphism; FM, fibromyalgia.

### 3.5. The Epigenetic Regulation of BDNF Expression in Chronic Pain

Chronic pain intricately involves abnormal gene expression within the neural cells responsible for processing nociceptive signals in the brain [152,153]. While genetic alterations offer a partial explanation for chronic pain, the emerging field of epigenetics provides a more nuanced and dynamic perspective by unraveling the gene expression patterns associated with chronic pain [154,155]. Recent studies revealed that epigenetic mechanisms, including histone acetylation [156], non-coding RNAs [156,157], and DNA methylation [127,158,159], can influence the expression of *BDNF* (Figure 2, Table 3). These epigenetic modifications may contribute to the pathogenesis and symptomatology of chronic pain.

Polli et al. [127] reported that patients with chronic fatigue syndrome and comorbid fibromyalgia exhibited lower DNA methylation levels in the *BDNF* gene compared to healthy individuals. DNA hypomethylation was associated with elevated serum BDNF expression, which in turn correlated with symptoms and hyperalgesia in these patients [127]. Additionally, Tao et al. [126] suggested that persistent inflammation could epigenetically upregulate *BDNF* protein expression, in turn increasing hypersensitivity and pain levels. Recent studies have shown that miRNAs can regulate *BDNF* expression and function. For example, findings from animal research indicate that miR-206-3p regulates *BDNF* expression through a conserved binding site in its 3′-UTR [160]. Notably, a study demonstrated that electroacupuncture alleviated neuropathic pain induced by chronic constriction injury by increasing miR-206-3p expression and inhibiting *BDNF* overexpression [161]. However, administration of the miR-206-3p inhibitor partially impaired the analgesic effect of electroacupuncture and the level of BDNF was elevated as well.

It is crucial to note that *BDNF* is highly susceptible to environmental influences (e.g., physical exercise, diet and nutrition, stress, and sleep patterns), which can, in turn, impact gene expression. For example, regular aerobic exercise has been associated with increased levels of BDNF [162], while chronic stress and elevated cortisol levels correlate with a reduction in *BDNF* expression [163]. Additionally, disrupted sleep patterns and chronic sleep deprivation can negatively impact BDNF levels [164]. From a biological perspective, an organism’s responses to the external environment are mirrored in epigenetic changes, influencing neuronal activity, such as cortical excitability and synaptic plasticity, thereby triggering behavioral alterations [165]. Moreover, alterations in chromatin structure represent one mechanism through which pain gradually transforms into the pathological processes of neuroinflammation, CS, and ultimately chronic pain [166].

## 4. Clinical and Methodological Implications

### 4.1. BDNF Treatment for Chronic Pain in a Broader Picture

*BDNF* serves as a driving force behind neuroplasticity in the context of chronic pain, positioning it as a potential biomarker and a novel therapeutic target. Although our understanding of *BDNF*’s role in pain processing remains limited, emerging evidence suggests its pro-nociceptive involvement in initiating and sustaining CS among individuals with persistent pain. Consequently, exploring the pharmacological and non-pharmacological manipulation of *BDNF* opens up crucial avenues for research. Various therapeutic strategies known to influence the release of BDNF have been extensively studied for regulating BDNF levels in patients with chronic pain, including neuromodulation techniques, *BDNF*-blocking therapies, and exercise therapy.

Neuromodulation techniques, such as transcranial direct current stimulation (tDCS), emerge as a promising treatment with analgesic properties [167,168,169]. By interfering with ongoing neural activity associated with pain processing and manipulating neuroplasticity and cortical excitability in specific brain regions, tDCS has been reported to improve pain and pain-related symptoms in patients with chronic pain [170,171]. While the exact mechanisms underlying these effects remain unclear, accumulating research suggests that the impact of tDCS may be neuroplasticity-state-dependent [172,173], with alterations in BDNF levels predicting the effects of tDCS on behavioral outcomes [174,175]. In other words, the analgesic effect of tDCS may depend on changes in endogenous *BDNF* levels [176,177], as *BDNF* is a driving force behind neuroplasticity [66]. A recent clinical study in patients with knee osteoarthritis revealed that active tDCS induced a significant reduction in both serum BDNF levels and pain intensity, compared to sham tDCS [178]. Consistent with previous preclinical studies, active tDCS reverted behavioral alterations associated with neuropathic pain, and decreased both serum and cerebral cortex BDNF levels [177]. Similarly, active tDCS alleviated nociceptive hypersensitivity induced by ovariectomy and reduced hippocampal BDNF levels [179]. These findings not only demonstrate the analgesic potential of tDCS but also suggest that tDCS-induced changes in neural circuits involved in pain processing correspond to alterations in BDNF levels, wherein reduced BDNF levels might directly contribute to pain relief. 

Exercise therapy seems to hold the capability to influence *BDNF* expression. Recent insights from a systematic review and meta-analysis within pain populations reveal an upregulation of *BDNF* expression in peripheral blood following diverse physical activities, accompanied by decreasing pain severity [180,181,182]. Similarly promising outcomes have been observed in healthy individuals [183]. However, the duration of exercise can yield varied results, with a single session or acute exercise reportedly increasing BDNF levels, while long-term or regular exercise may reduce them [184,185,186]. Notably, among neurotrophins, BDNF appears particularly responsive to exercise and physical activities [187]. Furthermore, the interplay of physical activity/inactivity extends to influencing epigenetic modifications, potentially inducing changes in *BDNF* expression [188,189].

In animal studies, the administration of a BDNF inhibitor or TrkB inhibitor was shown to reduce pain-like behavior. This suggests that *BDNF*-blocking therapies offer a viable therapeutic approach, targeting *BDNF* and/or its receptors through both pharmacological and non-pharmacological approaches. The TrkB receptor, serving as an endogenous receptor for BDNF and abundantly expressed in primary sensory neurons, is upregulated in chronic pain states, implicating BDNF/TrkB signaling in the process of CS [93,190]. TrkB-Fc, a chimeric compound sequestering endogenous BDNF, has shown promise in blocking *BDNF* effects on synaptic plasticity both in vivo and in neuropathic pain models [191]. Furthermore, spinal administration of TrkB-Fc has been reported to successfully reverse pain behavior in neuropathic pain models [192].

Medications can influence BDNF levels through various mechanisms, one of which is to reduce inflammation by targeting specific pro-inflammatory cytokines [193,194]. Steroids, commonly used to alleviate inflammation in immune-mediated diseases, have been demonstrated to inhibit *BDNF* expression in neurons [193]. While the pharmacological manipulation of *BDNF* expression holds promise for novel treatments, navigating the complex effects of different medications—often necessitating multiple drugs for patients—adds intricacy to the assessment of the final outcome. Notably, the potential adverse effects of blocking *BDNF* must be carefully considered, given its neuroprotective role in promoting neuronal growth and survival. Consequently, finely tuning *BDNF* expression, location, and secretion becomes imperative to fulfill its complex duties. A comprehensive understanding of these processes holds the key to developing new molecules and treatments for diseases associated with *BDNF*.

### 4.2. Using BDNF as an Objective Biomarker

Peripheral blood BDNF has been proposed as a potential biomarker related to disease activity and neuroprogression in various diseases [195,196,197], speculated to mirror alterations in brain expression of *BDNF*. This intricate relationship between brain and blood BDNF levels underscores the potential utility of peripheral measurements as informative markers for CNS dynamics. Given the challenges in directly measuring BDNF levels in the human brain, most clinical studies resort to using plasma or serum samples as proxies [47,198]. Measuring circulatory BDNF in peripheral blood, specifically in serum and plasma, provides a reliable and easily accessible method for sample collection with minimal disadvantages for patients [199].

*BDNF* polymorphisms have emerged as promising pain biomarkers. Specifically, the *BDNF* Val66Met polymorphism has been detected in diverse chronic pain populations, providing valuable insights into the susceptibility to distinct chronic pain conditions and the considerable interindividual variations in responses to various pain therapies [137,145,200]. Importantly, current literature suggests that *BDNF* polymorphisms can be reliably measured in both peripheral blood and buccal swab samples, making them accessible for potential diagnostic applications. Moreover, the development of tests to detect and define chronic pain conditions in the presence of the Val66Met polymorphism is an intriguing prospect. Several techniques, such as genotyping assays or real-time polymerase chain reaction (PCR) methods, could be explored to identify this specific genetic variant in blood samples. Additionally, advancements in genomic technologies, like next-generation sequencing, could provide a more comprehensive analysis of multiple genetic factors, including *BDNF* polymorphisms, in a single test. Despite these advancements, it is crucial to acknowledge the intricate nature of chronic pain, which results from a complex interplay of genetic, environmental, and psychological factors. While certain genetic variations may contribute to an individual’s susceptibility to pain or influence their responsiveness to pain treatments, it is improbable that a single genetic polymorphism, such as one related to *BDNF*, can conclusively pinpoint or define a chronic pain condition. Future research endeavors should strive to integrate different genetic risk factors with patient characteristics and clinical and psychological parameters to comprehensively address the multifaceted nature of chronic pain disorders.

The absence of biomarkers for diagnosing chronic pain remains a significant challenge in clinical practice. Typically, pain severity is assessed through the patient’s subjective report, an approach constrained by difficulties in quantification, reliability, and interparticipant comparability. The integration of objective biomarkers directly linked to the presence and severity of chronic pain would significantly (a) enhance the diagnosis and classification of pain pathophysiology, (b) assist with disease prognostication or predicting therapy responses, and (c) facilitate the development of innovative, mechanism-based treatment approaches, thereby reducing the reliance on long-term opioid use. Overall, *BDNF* is one of the most promising biomarkers for chronic pain disorders; however, a definitive clinical validation is still lacking.

## 5. Conclusions and Future Directions

The exploration of perpetuating factors in pain pathophysiology and their potential as therapeutic targets is crucial, given that existing treatments for patients with persistent pain, while effective, often fall short of expectations. CS is a form of maladaptive neuroplasticity underlying different chronic pain conditions and *BDNF* is a driving force behind neuroplasticity, influencing optimal brain function. This maladaptive neuroplasticity has been observed in the literature in subjects with similar characteristics. In this sense, it is possible to find therapies that promote the correct adaptation of this induced plasticity, thus providing a way to improve pain and functional recovery in individuals with persistent pain. Several studies have reported altered levels of both brain and peripheral BDNF in pain populations. However, the exact pathophysiological mechanisms driving these changes remain incompletely understood. The existing evidence does not conclusively determine whether alterations in BDNF levels are a cause or a consequence of chronic pain. Therefore, future research directions should prioritize elucidating the multifaceted role of *BDNF* in chronic pain, particularly given its nuanced actions dependent on pain type, site of expression/secretion, receptor type, and gene polymorphisms. This review also delves into *BDNF* polymorphisms and epigenetic regulation of *BDNF* expression within the context of chronic pain. Currently, most evidence comes from cross-sectional and preclinical studies. To comprehensively understand the dynamics of *BDNF* expression via epigenetic regulation in various chronic pain conditions, further research with larger sample sizes and longitudinal studies in pain populations is imperative. This approach will facilitate a more in-depth exploration of the intricate interplay among genetics, epigenetic modifications, and *BDNF* in the development and persistence of chronic pain.

## Figures and Tables

**Figure 1 biomolecules-14-00071-f001:**
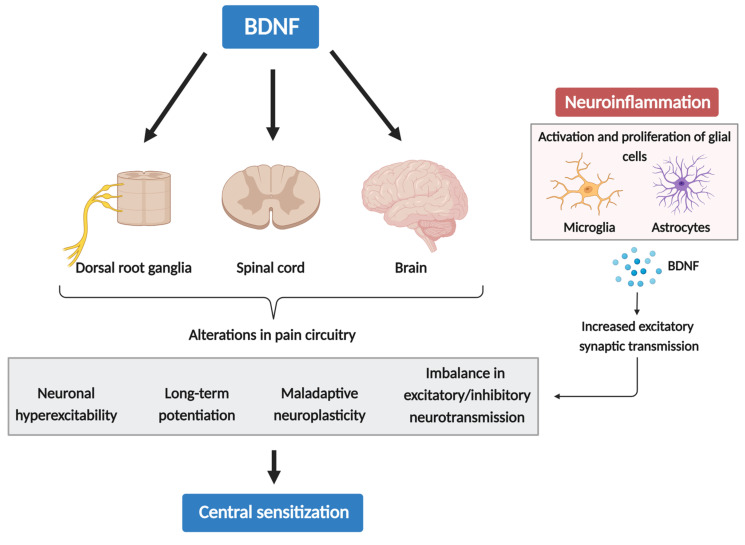
The role of BDNF in central sensitization and neuroinflammation.

**Figure 2 biomolecules-14-00071-f002:**
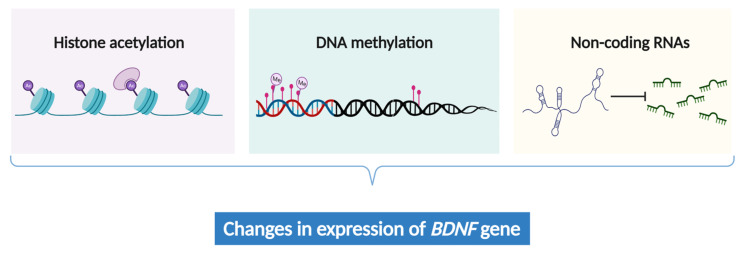
The epigenetic regulation of *BDNF* expression.

**Table 3 biomolecules-14-00071-t003:** DNA Methylation Changes in *BDNF* gene in patients with chronic pain.

Reference	Study Population	Study Design	Epigenetic Assessment	Tissue	Assay	Main Results
Polli et al., 2020 [127]	Chronic fatigue syndrome and comorbid fibromyalgia (n = 28) Healthy controls (n = 26)	Cross-sectional	DNA methylation (*BDNF* gene-specific)	Blood	PCR amplification, pyrosequencing	Compared to controls, serum BNDF was higher in patients with CFS/FM (mean difference of 3.31 ng/mL; *p* = 0.001), whereas *BDNF* DNA methylation in exon 9 was lower (mean difference of −2.16%; *p* = 0.007). Lower methylation in the same region predicted higher BDNF levels (*p* = 0.009), which in turn predicted participants’ symptoms (*p* = 0.001) and widespread hyperalgesia (*p* = 0.044).
Paoloni-Giacobino et al., 2020 [158]	Chronic musculoskeletal pain (n = 58) Healthy controls (n = 18)	Cross-sectional	DNA methylation (*BDNF* gene-specific)	Blood	PCR amplification, pyrosequencing	The methylation values of *BDNF* were significantly (*p*< 0.005) increased 1.9-fold in patients with CMS as compared with the healthy controls. A high level of biopsychosocial complexity was associated with lower average CpG methylation values of *BDNF* (*p* = 0.02) in patients with CMS, and may therefore increase the level of BDNF. The upregulation of *BDNF* is associated with higher levels of biopsychosocial complexity.
Menzies et al., 2013 [159]	Fibromyalgia (n = 10) Healthy controls (n = 8)	Cross-sectional	DNA methylation (genome-wide)	Blood	450 K human methylation assay	Authors found 69 differently methylated positions, in 47 different genes, i.e., *AXL*, *HDAC4*, *BDNF*, *PRKCA*, *RTN1*, *PRKG1*, *SOD3*. There is a significant difference in the methylation pattern of the *BDNF* gene between patients with FM and controls.

BDNF, brain-derived neurotrophic factor; PCR, polymerase chain reaction; CMS, patients with chronic musculoskeletal pain; CFS, chronic fatigue syndrome; FM, fibromyalgia.

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
