# Peer review of "The Role of the Brain-Derived Neurotrophic Factor in Chronic Pain: Links to Central Sensitization and Neuroinflammation"

_biomolecules, 2024, doi:10.3390/biom14010071_

Round 1

Reviewer 1 Report

Comments and Suggestions for Authors

This review is devoted to the role of the brain-derived neurotrophic factor in chronic pain: links to central sensitization and neuroinflammation. Chronic pain is sustained, in part, through the intricate process of central sensitization, marked by maladaptive neuroplasticity and neuronal hyperexcitability within central pain pathways. Accumulating evidence suggests that CS is also driven by neuroinflammation in the peripheral and central nervous system. In any chronic disease, the search for perpetuating factors is crucial in identifying therapeutic targets and developing primary preventive strategies. Comprehensively understand the profound impact of BDNF in chronic pain, authors delve into its key characteristics, focusing on its role in underlying molecular mechanisms contributing to chronic pain. Authors also explore the potential utility of BDNF as an objective biomarker for chronic pain. This discussion encompasses emerging therapeutic approaches aimed at modulating BDNF expression, offering insights into addressing the intricate complexities of chronic pain.

However, in my opinion, the authors' claim that the BDNF cross the blood-brain barrier freely is controversial and relies on quite old publications and needs correction.

Author Response

Dear reviewer,

Thank you very much for taking the time to review this manuscript.

We appreciate your positive feedback and the constructive suggestion, and hope that the revised manuscript meets your expectations.

Please find the detailed responses below and the corresponding revisions/corrections highlighted in the attached revised files.

  1. However, in my opinion, the authors' claim that the BDNF cross the blood-brain barrier freely is controversial and relies on quite old publications and needs correction.

Response:

In light of your suggestion, we have corrected related information and updated latest references. (Page 3, Line 100-109)

In addition, circulating BDNF derives from both peripheral and cerebral sources [39-41]. Notably, in human studies, serum [42-44] and plasma [45] BDNF levels appear to correlate positively with BDNF levels in the brain. Moreover, findings from animal studies suggests a positive correlation between peripheral blood BDNF levels and its concentration in the cerebral cortex, indicating that fluctuations in peripheral blood BDNF levels may reflect changes in the brain [46-50]. However, recent studies comparing cerebrospinal fluid and serum BDNF measurements in patients with Alzheimer’s disease have indicated a lack of correlation [51] and that the levels of BDNF in serum or plasma were found to be significantly higher than those in cerebrospinal fluid, possibly due to peripheral synthesis [52]. 

Reviewer 2 Report

Comments and Suggestions for Authors

The review focus its attention on BDNF, adhering at the requirement from the editorial department for the Special Issue of Biomolecules. The authors explore the role of BDNF in regulating the brain plasticity, or central sensitization (CS), in response to chronic pain, a condition that affect millions of people worldwide.

In this article, in particular, they report how BDNF is generate and operates in the nervous system via the TrkB and p75NTR receptors, mentioning also the molecular pathways downstream of them. The authors highlight how BDNF has several functions in the context of pain by affecting neuroplasticity and neuronal excitability. It is clearly highlighted how a fine tuning of BDNF levels is required for optimal functions of the nervous system. 

The article per se fits all the requirement for the publication in your journal, though some formatting modification may be required.

Specifically, the authors reported conflicting findings on the role of BDNF as pro-/anti-nociceptor, though I feel like this information is lost in the following sections regarding the genetic and epigenetic controls of BDNF expression in pain related conditions. Since a lot of information is reported in this section, I would suggest to change the figure 2 into a table with clearly reported modification of BDNF at genetic/epigenetic/protein level that are associated with painful condition or synaptic plasticity. A table or a figure highlighting the contradicting effect of high levels of BDNF in the brain may also be useful to immediately capture the fascinating complexity of the role of this protein.

The duality of BDNF previously reported is also lost in the section where clinical implications are explored. Moreover, we can find in this paragraphs additional information on the basic epigenetic regulation of the protein (specifically on miR-30a, miR-206) , a concept that was already explored before and with limited implication on the clinical aspects.  

On the biomarker section, BDNF is presented as a good possible biomarker due to its presence in the circulating blood. However, from the text it is unclear if the Val66Met polymorphism mentioned is also found in blood samples. I feel like this part could be expanded in speculation on which type of test could be developed to detect/define a chronic pain condition in the presence of this polymorphism. 

Overall, the review describes an interesting role of BDNF in modulating nociception and open the road to speculation regarding its pharmacological targeting to achieve analgesia. 

Comments on the Quality of English Language

The grammar is fine, some concepts are repetitive can be improved.

Author Response

Dear reviewer,

We appreciate your positive feedback and the constructive suggestions, and hope that the revised manuscript meets your expectations.

Please find the detailed responses below and the corresponding revisions/corrections highlighted in the attached revised manuscript.

  1. Specifically, the authors reported conflicting findings on the role of BDNF as pro-/anti-nociceptor, though I feel like this information is lost in the following sections regarding the genetic and epigenetic controls of BDNF expression in pain related conditions. Since a lot of information is reported in this section, I would suggest to change the figure 2 into a table with clearly reported modification of BDNF at genetic/epigenetic/protein level that are associated with painful condition or synaptic plasticity. A table or a figure highlighting the contradicting effect of high levels of BDNF in the brain may also be useful to immediately capture the fascinating complexity of the role of this protein.

Response:

Thank you for your great suggestions. In light of your suggestion, we have added three tables to report the modification of BDNF at genetic/epigenetic/protein level in patients with chronic pain.

Table 1. BDNF levels and association with pain in patients with chronic pain.

Table 2. BDNF Val66Met Polymorphism in patients with chronic pain.

Table 3. DNA Methylation Changes of BDNF gene in patients with chronic pain

  1. The duality of BDNF previously reported is also lost in the section where clinical implications are explored. Moreover, we can find in this paragraphs additional information on the basic epigenetic regulation of the protein (specifically on miR-30a, miR-206), a concept that was already explored before and with limited implication on the clinical aspects.

Response:

We have modified the content about the duality of BDNF in this section. (line 383-388). Additionally, we also removed the repetitive information.

Notably, the potential adverse effects of blocking BDNF must be carefully considered, given its neuroprotective role in promoting neuronal growth and survival. Consequently, finely tuning BDNF expression, location, and secretion becomes imperative to fulfill its complex duties. A comprehensive understanding of these processes holds the key to developing new molecules and treatments for diseases associated with BDNF.

  1. On the biomarker section, BDNF is presented as a good possible biomarker due to its presence in the circulating blood. However, from the text it is unclear if the Val66Met polymorphism mentioned is also found in blood samples. I feel like this part could be expanded in speculation on which type of test could be developed to detect/define a chronic pain condition in the presence of this polymorphism.

Response:

Current literature suggests that BDNF polymorphisms can be reliable measured in both peripheral blood and buccal swab samples (see table 2), making them accessible for potential diagnostic applications.

Moreover, in light of your suggestion, we have expanded related information about BDNF polymorphisms in section 4.2 “Using BDNF as an objective biomarker”. (line 399-419)

BDNF polymorphisms have emerged as promising pain biomarkers. Specifically, the BDNF Val66Met polymorphism has been detected in diverse chronic pain populations, providing valuable insights into the susceptibility to distinct chronic pain conditions and the considerable interindividual variations in responses to various pain therapies [137,145,200]. Importantly, current literature suggests that BDNF polymorphisms can be reliable measured in both peripheral blood and buccal swab samples, making them accessible for potential diagnostic applications. Moreover, the development of tests to detect and define chronic pain conditions in the presence of the Val66Met polymorphism is an intriguing prospect. Several techniques, such as genotyping assays or real-time polymerase chain reaction (PCR) methods, could be explored to identify this specific genetic variant in blood samples. Additionally, advancements in genomic technologies, like next-generation sequencing, could provide a more comprehensive analysis of multiple genetic factors, including BDNF polymorphisms, in a single test. Despite these advancements, it is crucial to acknowledge the intricate nature of chronic pain, which results from a complex interplay of genetic, environmental, and psychological factors. While certain genetic variations may contribute to an individual's susceptibility to pain or influence their responsiveness to pain treatments, it is improbable that a single genetic polymorphism, such as one related to BDNF, can conclusively pinpoint or define a chronic pain condition. Future research endeavors should strive to integrate different genetic risk fac-tors with patient characteristics, clinical and psychological parameters to comprehensively address the multifaceted nature of chronic pain disorders.